# The Relationship between Personality Traits and Facebook Addiction among Adolescents in an Urban, Rural and Semi-Rural Secondary School

**DOI:** 10.3390/ijerph192013365

**Published:** 2022-10-16

**Authors:** Mokoena Patronella Maepa, Alicia Wheeler

**Affiliations:** 1Clinical Psychology Department, School of Medicine, Sefako Makgatho Health Sciences University, Ga-Rankuwa 0208, South Africa; 2Psychology Department, Health Sciences Faculty, North-West University, Mafikeng 2735, South Africa

**Keywords:** social networking sites, Facebook, Facebook addiction, personality traits

## Abstract

Facebook is and was intended to provide a place for friends to connect within the bustling academic environment and to encourage openness for ideas and interests. When used sparingly, it can provide an individual with a sense of group belonging and connection, sharing and offering hope and advice. The misuse of Facebook can have detrimental effects on one’s quality of life that often lead to addiction. In this correlation design study, secondary-school-aged adolescents’ Facebook addiction was compared to personality attributes. Through a convenience sample, 240 teenagers in total (106 men and 134 females) were chosen. The respondents answered questions about their demographics, Facebook Addiction, and Junior Eysenck Personality. The findings showed a substantial inverse correlation between Facebook addiction and neuroticism (r = −0.260, *p* < 0.01) and psychoticism (r = −0.189, *p* < 0.01). There was no discernible statistical link between Facebook Addiction and Extraversion. The study comes to the conclusion that although social networking sites such as Facebook have good effects on adolescents’ lives, their use needs to be regulated, the risks were highlighted, and at-risk individuals can receive intervention approaches, such as social skills training.

## 1. Introduction

Mark Zuckerberg founded Facebook in February 2004 as a social network for Harvard students [1]. Facebook serves as a tool for app development, enabling the creation of fresh, cutting-edge features, as well as independent goods and services [1]; being a well-liked platform, Facebook promoted the acquisition of gadgets and internet connections, increasing connectivity [2]. While there are many social media platforms, such as WhatsApp, Instagram, Tiktok, Snapchat, and Twitter, Facebook is the largest globally, with 1.82 billion daily active users as of the third quarter of 2020 [2]. The global FB demographic data of 2020 show that young adults aged 18–34 account for 55 percent of the total users, of which 4 billion are active users on Facebook daily [2]. With the availability of Facebook Free Mode, users can access the site without being connected to the internet, making it the most cost-effective accessible social media site.

Humans are social creatures who interact and communicate with other people all the time. People are looking for novel ways to stay in touch with one another in today’s technologically evolved society. For this purpose, social networking is a daily activity for people all over the world who are linked to it. According to Griffiths [3], social networking sites (SNSs) are online communities that include websites such as Facebook, Twitter, and Instagram. Facebook, which is referred to as an interactive, web-based network, is the subject of this study [4].

Among all other social media platforms, Facebook is one of the most used social network sites worldwide [5]. At the worldwide level, there are more than 2.27 billion monthly active users and 1,15 billion daily active users [5]. Currently, there are approximately 2.4 million Facebook users in South Africa who visit the site more than three times every day. Of these users, 76 percent are men, and 24 percent are women [5]. By 2026, there will likely be 34.92 million active Facebook users. According to NapoleonCat [6], there were 25,450,000 Facebook users in South Africa in May 2021, or 42.3 percent of the country’s total population. Women made up 50.1 percent of Facebook users, and users between the ages of 25 and 34 made up the largest user demographic [6]. Facebook users, on average, spend 60 min per day on the social media platform and log in three to five times each day [7].

Facebook provides people with the chance to communicate with others about a huge variety of topics by sharing information and ideas. Additionally, technology makes it possible for people to communicate with others who share their interests regardless of political, economic, or geographic boundaries [8]. Donnelly [9] and Wang, Kang, Tsai, Song, and Lien [10], among others, found that the excessive use of social networks causes a behavioral addiction to internet sites, even while these connections or relational benefits are acknowledged. In addition, these researchers discovered that Facebook users spend 15 h a month on its mobile app, which might cause addiction and mental health issues. More research is needed to find ways to lessen or eliminate Facebook usage and addiction. This is so because there are risks associated with social media use; unfortunately, some studies opt to report social media benefits, ignoring the risks [11]. Such risks are demonstrated by Imani et al. [12], who indicated that there was an association between social media use and quality of life among people with depression and anxiety.

Physical, social, and emotional development, as well as other areas, all take place during adolescence [13]. Social growth, which is linked to social identity, also occurs during this time [14]. Healthy social development allows people to build strong bonds with their family, friends, instructors, and other important people in their lives [15]. It also helps kids develop moral judgment and values, boosts their self-esteem, and gives them information that differs from what they have learned at home [14]. Teenagers will spend an increasing amount of time on Facebook in order to receive this feedback [16]. Technology now plays a significant influence in many aspects of our everyday lives, including the social development of adolescents [13]. Teenagers could be at risk of being addicted to social networks such as Facebook [17]. In this regard, the internet is crucial for adolescent identity exploration, intimacy and a sense of belonging, separation from parents and family, as well as for expressing their displeasure and success [18]. Fitting in with friends is one of the most essential things in life during this time in a person’s growth, and Facebook provides them with the tools and knowledge they need to do just that.

According to Erikson [19], a person needs to master the identity versus role confusion stage during adolescence in order to develop their personality. During this phase, adolescents experience a period of confusion where they must explore, question their existing beliefs, and try out alternative roles in order to eventually come up with their own values and beliefs [13]. Because this experimenting typically occurs in a social setting, how teenagers interact with others has a significant impact on how they develop as people [20].

According to the American Psychiatric Association [21], personality is a collection of organized psychological qualities and systems that are present in each individual. Extraversion, neuroticism, openness to experience, consciousness, and agreeableness are among the personality traits [22]. These characteristics and systems persist and affect how people interact with others and adapt to their environment [21].

Individuals with particular personality qualities, such as extraversion and neuroticism, are more likely than others to develop a Facebook addiction [23]. These characteristics are described by Larsen and Buss [24] as a collection of psychological qualities and systems that are organized within the person and reasonably durable. This affects how they interact with the environment and how they adapt to it. According to Sherman [23], those with neurotic personality qualities are more prone to develop a Facebook addiction than people who exhibit extroversion and an openness to new experiences. This may be due to the fact that people with neurotic personality traits have more trouble interacting with others in person [23]. However, these individuals use other strategies, such as Facebook, to feel as though they belong since they feel the urge to be a part of a particular group during adolescence. According to other research by Kuss and Griffith [17] and Malik and Khan [25], youth with high extraversion and low conscientiousness are more prone to become hooked to these social networks, while those with neurotic personality traits are more likely to do so. This indicates that people who exhibit extrovert and neurotic personality qualities are more prone to develop a Facebook addiction. The use of Facebook is also associated with receiving feedback. This feedback-seeking on social media is associated with a focus on self-presentation [26]. This is more commonly reported among females who use social media engagements with the intention of satisfying their social needs [27].

The structure of online communities has changed along with the popularity of SNS. Social networking sites are noticeably organized around people and less by common interests than websites that are only dedicated to their communities of interest [28]. Therefore, it would make sense to concentrate future SNS research on users of social networking sites. People, especially teenagers, appear to spend a lot of time on Facebook. According to World Wide Worx and Fuseware [29], Facebook use and browsing have become significant social communication behaviors. As a result, recent social network research has focused on how people feel increasingly pressured to keep up with their online relationships and how people use social media to gain popularity. This leads to an obsession with social media and excessive use of SNSs [30,31]. Facebook addiction has not received the necessary attention, according to Xu and Tan [32], despite the fact that people spend an increasing amount of time on social media.

Researchers Kuss and Griffiths [17], Sherman [23], Smith-Duff [33], and Malik and Khan [25] investigated Facebook Addiction and its associations with personality factors in regard to this assertion. However, there has not been enough research undertaken in the South African context, where various cultures may have diverse opinions on Facebook. However, social media is available to teenagers everywhere, and each time a user logs in, the chance of developing a Facebook addiction rises [34]. By adding information about how specific personality factors can increase a person’s likelihood of developing Facebook addiction, this study will fill the knowledge vacuum in the area of Facebook addiction research. As a result, this study investigates the connection between adolescent personality traits and Facebook addiction, assuming that there will be a statistically significant relationship between the two.

Internet addiction, according to Widyanto and Griffiths [35], produces notable issues in both social and professional life. These issues include mental health issues such as anxiety, negative personal development issues, detrimental effects such as withdrawal and over-engagement, serious disorders such as friendship difficulties, depression, poor sleep, excessive mental activity, and recurrent thoughts about controlling use and failure to prevent access [31]. The study aims to explore the relationship between personality traits and Facebook addiction among secondary school adolescents in rural settings.

## 2. Methods

### 2.1. Design

In order to examine the connection between Facebook addiction and personality traits, a correlation design was used in this study, which used a quantitative research approach.

### 2.2. Sample

Overall, 240 adolescents from a secondary school in Pretoria were chosen for the study using a convenience sample approach; 106 (44.1 percent) were male, and 134 (55.9 percent) were female. Their ages ranged from 13 to 20 years.

### 2.3. Instruments

#### 2.3.1. Facebook Addiction

The Bergen Facebook Addiction Scale was used to measure Facebook addiction (BFAS). According to Torsheim, Brunborg, and Pallesen [36], the original BFAS consisted of 18 items, one for each of the six main components of addiction: salience, mood modulation, tolerance, withdrawal, conflict, and relapse. There are six items on the final scale because only the item from each of the six addictive aspects with the highest adjusted item-total correlation was maintained. Each response is graded on a 5-point scale, with one representing “Very rarely” and 5 representing “Very regularly”. With a cut-off score of at least 12 for Facebook Addiction, higher scores imply stronger Facebook addiction. A good factor structure and a coefficient alpha of 0.83 were discovered by the developer [37].

#### 2.3.2. Personality Traits

For the purpose of this study, Eysenck’s personality questionnaire will be used. The questionnaire takes both nature and nurture into account [38]. Four potential benefits of the questionnaire are (a) improved reliability, (b) improved validity, (c) better acceptance on the part of the people being evaluated, and (d) a clearer dimensional structure of the questionnaires [38]. Furthermore, the persons examined prefer the Likert-type format since it gives them more freedom in their response and does not limit them to a choice between just two options [38].

The Junior Eysenck Personality Questionaire—Revised Short Form—was used to measure the adolescents’ personality traits. This scale measures three pervasive, independent dimensions of personality, Extraversion-Introversion and Neuroticism-Stability, and Psychoticism, which account for most of the variance in the personality domain. Each questionnaire contains 48 “Yes-No” items with no repetition of items. According to Sato [9], the JEPQ-R Short Scale has good internal consistency, test–retest reliability, and concurrent validity in a variety of cultures. Reliabilities include the following for males and females, respectively, 0.84 and 0.80 for neuroticism, 0.88 and 0.84 for extraversion, 0.62 and 0.61 for Psychoticism, and 0.77 and 0.73 for the lie scale [39]. Inter-Item Correlation Matrices were employed in the current study to determine the scale’s reliability. The following Cronbach Alpha ratings were obtained for the various personality traits: 0.64 for extraversion, 0.60 for neuroticism, and.49 for psychoticism. It was discovered that item 30 on the Psychoticism scale did not have excellent inter-item reliability; when this item was eliminated, the Cronbach Alpha for the Psychoticism items within the scale was 0.51. These findings suggest that, with the exception of a few questions pertaining to Psychoticism personality traits, the test exhibited strong intra-item reliability.

### 2.4. Procedure

Informed consent forms were distributed to the parents of the learners, and assent forms for the learners who were under the age of 18 after the study received ethical clearance from the North-West University Ethics Committee (ethical clearance number: NWU-00340-17-A9). The Department of Education and the school also gave their further approval for the study. The participants were informed of the procedure after completing the consent and assent forms.

The teachers of each class and the researcher then delivered the questionnaires to all the grades and explained them to the participants who had been chosen. Although the researcher was not present in the classes when the data were being collected, the teachers were there and available to answer any questions. The teachers were also given the researcher’s phone number in case there was a miscommunication. The Eysenck personality test and the Bergen Facebook Addiction Scale were used to ask the participants questions about their personality features and whether they had a Facebook addiction. The data were captured and analyzed after it had been collected.

### 2.5. Statistical Methods

The Statistical Package for the Social Sciences (SPSS 22), version 22, was used to analyze the data. The correlation coefficient was calculated to demonstrate the degree of relationship between the two variables (personality traits and Facebook addiction). In stated in Section 5, the findings are displayed as tables. The *t*-test was used to test hypothesis 2 (that females are more likely to be addicted to Facebook than males), and the Pearson-R correlation was used to test hypothesis 1 (that there will be a relationship between Facebook addiction and personality factors) (that older adolescents are more likely to be addicted to Facebook than younger adolescents).

## 3. Results

Table 1 represents the results of the Facebook addiction and personality trait scales that were administered.

The findings showed a substantial negative correlation between Facebook addiction and the following personality traits: psychoticism and neuroticism (r = −0.189, *p* < 0.01) and neuroticism and psychoticism (r = −0.260, *p* < 0.01). The results did show a minor association between the opposing pole of Extraversion and Facebook Addiction, even if there was no substantial relationship between Facebook Addiction and Extraversion (Facebook Addiction and Introversion).

## 4. Discussion

According to the statistical investigation, there is a connection between personality traits and Facebook addiction. According to McLeod [40], Eysenck’s personality traits—Extraversion—Introversion, Neuroticism—Stability, and Psychoticism—are characterized in terms of polar opposites. These personality dimensions and their second-order personality traits can also be used to explain the study’s findings. When Facebook Addiction and a personality attribute have a high negative correlation, it means the person has characteristics from the main personality dimension (Extroversion, Neuroticism, and Psychoticism). If a person scored higher on the extraversion–introversion scale, for instance, that person is probably more introverted. They are more likely to exhibit extraversion personality traits if they have a lower score.

The findings showed a strong inverse link between neuroticism and Facebook Addiction. This translates into lower neuroticism subtest scores for people who scored highly on the Facebook Addiction Questionnaire, which is a good indication of neuroticism and a negative indication of a stable personality. Therefore, persons who exhibit Neuroticism Personality Traits are more likely to develop a Facebook addiction. This may be due to the fact that people with neurotic personality traits frequently experience sudden spikes in anxiety and mood without any warning [40]. These teenagers may find it challenging to communicate with their peers in person as a result of having neurotic personality traits [41].

Eysenck [42] defined people with neurotic tendencies as those who are mostly stressed out, worried, pessimistic, afraid, and have low self-esteem. These people can find a secure space on Facebook where they are in charge. These young people can build profiles on Facebook that only reflect their desired image of themselves rather than their true selves. They experience less anxiety as a result, which leads to increased Facebook use and a higher risk of Facebook addiction [17]. These outcomes are consistent with the social network usage groups identified elsewhere. People who fall under the category of attention seekers all have traits of neuroticism, such as the need to be liked by others. The findings of this study concur with those made by Kuss and Griffiths [17], Mahmood and Farooq [41], and Omar and Subramanian [16].

The findings also showed a strong inverse association between psychoticism and Facebook Addiction. This implies a positive indication of Psychoticism and a negative indication of Conscientiousness for those who performed well on the Facebook Addiction subtest but had lower scores on the Psychoticism subtest. Therefore, those who possess Psychoticism Personality Traits are more likely to develop a Facebook addiction. These young people who exhibit increased psychoticism tend to be “loners” and dislike social interaction [43]. Additionally, they are characterized as selfish, cold, impulsive, innovative, and tough-minded [43].

Psychoticism is a personality trait that, according to Eysenck [42], describes people who take risks, are impulsive and are emotionally callous. It may seem weird that there is such a strong correlation between psychoticism and Facebook addiction because people with this personality trait tend to struggle with interpersonal relationships. One would argue that since they do not value emotional connections, they are likely to compensate for such through social media, making this peculiar. However, Chidi et al. [43] also found that they are egotistical, impulsive, and creative. They have a platform to creatively remake themselves, thanks to Facebook. Additionally, it offers a location where users may post details and images of themselves in an effort to make other people envious. Facebook thus provides a platform for their egocentric tendencies. They communicate with people on social networks instead of in person, which increases the amount of time they spend on Facebook and the risk of Facebook addiction. Studies carried out by Kuss and Griffiths [17], Mahmood and Farooq [41], as well as Blanchinio and Przepiorka [44] support these findings. According to these findings, those who have high Consciousness and low Psychoticism are less prone to become dependent on social networks such as Facebook.

The results did show a minor association between the opposing pole of Extraversion and Facebook Addiction, even if there was no substantial relationship between Facebook Addiction and Extraversion (Facebook Addiction and Introversion). This indicates that extroverted people are less prone to develop a Facebook addiction than introverted people. Extroverted people are characterized as outgoing, sociable, chatty, and vivacious people who prefer big social circles. People that are more introverted are perceived as timid, quiet, reserved, and lacking in confidence [42]. Thus, it may make more sense that extroverts are more likely to develop a Facebook addiction. Extroverts are more outgoing and love communicating with their peers, but they do both in person and online through social networks such as Facebook. However, introverted teenagers are more hesitant when speaking to their classmates face-to-face. They typically engage in less face-to-face interaction [40]. Facebook offers them a setting where they may socialize with their peers without feeling threatened. They are more likely to develop a Facebook addiction because this is now their preferred means of communication than people who communicate both face-to-face and through Facebook [41].

This is in line with the study findings by Mahmood and Farooq [41]. However, Blanchnio and Przepioka [44] discovered a connection between Facebook Addiction and extroverted people. The study by Andreassen et al. [45] also revealed this. As can be observed from the findings, there is unquestionably a connection between personality traits such as neuroticism, psychoticism, and introversion, as well as Facebook addiction. Adolescents go through a stage of identity and role confusion throughout this time [21]. Adolescents communicate largely with their peers throughout this stage in order to develop their self-expression skills and discover who they are and what they want to achieve with their lives [13]. Adolescents may act recklessly in this era because they feel untouchable and appear careless and selfish [13]. The development of independence, individuality, and self-esteem is another top priority during this time [46].

Several of these qualities that are unique to adolescents already raise their chance of becoming a Facebook addict. Thus, when these features are combined with other personality traits, such as introversion, neuroticism, and psychoticism, the likelihood that they may develop an addiction increases even further. For instance, a teenager with elements of neuroticism is a particularly worried person when it comes to social interactions. However, because the person is still a teenager, they still need to participate in social interactions with their classmates. Facebook accepts this with little concern. As a result, individuals use Facebook more frequently, and eventually, they are unable to interact with people outside of online networks. Thus, the adolescent stage emphasizes personality features that make people more susceptible to developing a social media addiction.

The study, therefore, has implications. These findings imply that general adolescent social development should be considered in the context of the effect of social media on their social skills and communication skills. We need to ask if this social media addiction mirrors alcohol or substance use addiction patterns. In addition to alcohol and substance use addictions, the negative impact of social media addiction should be explored. The findings further suggest that clinicians working with the adult population should consider the effect of social media on adolescent mental health challenges. Parents need to be actively engaged in monitoring and supervising adolescent use of social media to prevent exposing them to unwarranted age-inappropriate content. There need to be educational campaigns to educate youth on safe skills for social media use. There should be clear policies and guidelines governing adolescent social media use other than focusing only on age limits.

## 5. Conclusions

This study looked at how personality traits and Facebook addiction correlated among teenagers from a high school in the Mafikeng area. According to the findings, there is a substantial correlation between a few personality traits and Facebook addiction, particularly neuroticism and psychoticism. More introverted people appear to be more susceptible to developing a Facebook addiction. However, there are certain advantages to using social networking services. Social networking sites can improve young people’s media literacy by teaching them how to use the internet safely and to their benefit. Additionally, it might foster creativity, the growth of individual identities, self-expression, social connections, a sense of belonging, and community building (Collin et al., 2011). However, this does need to be improved by other people, such as teachers and parents, and this will be covered in more detail in the proposals.

## 6. Limitation

There are a few limitations to this study. First off, only one school in the Pretoria district was used for the study. One would need to expand the study to more than one high school in the area in order to gather more useful data that covers a broader sample of teenagers. Additionally, the diverse ethnic backgrounds of the teenagers were not a focus of the current study. The socioeconomic status of teenagers and their familiarity with technology were not examined as variables. These factors might have affected the adolescents’ access to social media, which would have affected the study’s findings.

Additionally, there was no control group that included teenagers who use Twitter, Instagram, or WhatsApp. Some teenagers may be dependent on a different social network, or they may use many social media platforms in addition to Facebook. Therefore, they would be addicted to social media in general rather than Facebook particularly. A lack of prior studies on Facebook addiction was another drawback, which reduced the amount of material that could be included. The current study should contribute to the growing body of research on social networks, the Internet, and the rising prevalence of addiction among teenagers.

## Figures and Tables

**Table 1 ijerph-19-13365-t001:** Correlation Matrices for Facebook Addiction and Personality Traits.

Variables	FA	EX	PS	NE
Facebook Addiction (FA)	1			
Extraversion (EX)	0.001	1		
Psychoticism (PS)	−0.189 **	0.066	1	
Neuroticism (NE)	−0.260 **	−0.095	0.217 **	1

** Correlation is significant at the 0.01 level (2-tailed). Note: FA = Facebook Addiction, EX = Extraversion, PS = Psychoticism, NE = Neuroticism.

## Data Availability

The datasets generated for this study are available on request from the corresponding author.

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
