# Peer review of "The Relationship between Personality Traits and Facebook Addiction among Adolescents in an Urban, Rural and Semi-Rural Secondary School"

_ijerph, 2022, doi:10.3390/ijerph192013365_

Round 1

Reviewer 1 Report

Line No.

26 - Facebook name has been changed to Meta and has new guidelines.

38 - Unfortunately content may be censored because fact (opinion) checkers are ether biased politically or uninformed.  Need to have a qualifier indicating this may happen.

30 - and throughout the paper.  When presenting percentages in text, write percent instead of using the symbol which are okay in tables and charts.

78 - Should that be the American Psychological Association?

Maybe there should be a warning posted by Meta (FB) of the dangers of addiction and sleep deprivation and other anomalies.  But the sleep problem may be the result of the blue light emanating form the computer.

146 - single digits should be written, e.g., one instead of 1. The hyphen with 5-point makes the single digit an exception.

186 - Move SPSS-22 to just after the full title.

195 - Need introduction to tables regarding purpose and and brief summaries after the tables of what was found.

257 - Do not use contractions in formal writing.

297 - better to use several rather than numerous ...

321 - limitations and delimitations.  There are both characterized int the study.  You should explain both.

If you italicize journal titles, it makes it easier to recognize the journal.

Author Response

See the attached file for the editions done on the comments

Reviewer 2 Report

Dear authors, thank you very much for your submission to IJERPH and for giving me the opportunity to review your paper on which traits foster Facebook addiction. The topic is suitable for the journal’s aims and scope. Your research question is fundamental and worth investigating. Your paper is well-structured and well-written. You provide almost all necessary information. In the following, I report on several issues that should be addressed to further increase the manuscript quality:

Title

  1. I would recommend to avoid brackets in a title.

Abstract

  1. Please start your abstract with a sentence about the relevance of your research. Why is FB addiction a problem?
  2. Please briefly mention the good effects FB may have on adolescents (“such as…”).

Introduction

  1. To my knowledge, FB is no longer the leading platform for secondary school students. It’s rather TikTok and Instagram, maybe Snapchat. Could you please briefly justify why you focus on FB?
  2. Please clearly state your research question(s) or goal(s).

Methodology

  1. Please briefly explain why you chose the Eysenck trait system rather that the more common Big Five approach.

Discussion

  1. Please discuss the theoretical, practical, and policy implications of your findings.

Limitations and further research

  1. As this section is very short, think about making this section 6.4.

References

  1. In my opinion, it is always advisable to show a high knowledge about the ongoing discussion in your target journal. Therefore, I recommend to add a few more recent (last three years) articles from IJERPH.
  2. For your introduction, please check, if these references may be helpful: doi:10.3390/fi12090146, doi:10.1007/s12144-018-9998-0, doi:10.2196/27000.

Thank you for your contribution! I hope you find my comments helpful. Good luck with your revision!

Author Response

see attached documents for review comments made

Round 2

Reviewer 2 Report

Thank you for your revision and for addressing my comments!